# Protein Intake and Protein Quality Patterns in New Zealand Vegan Diets: An Observational Analysis Using Dynamic Time Warping

**DOI:** 10.3390/nu17111806

**Published:** 2025-05-26

**Authors:** Bi Xue Patricia Soh, Matthieu Vignes, Nick W. Smith, Pamela R. von Hurst, Warren C. McNabb

**Affiliations:** 1Sustainable Nutrition Initiative, Riddet Institute, Massey University, Palmerston North 4410, New Zealand; m.vignes@massey.ac.nz (M.V.); n.w.smith@massey.ac.nz (N.W.S.); w.mcnabb@massey.ac.nz (W.C.M.); 2School of Mathematical and Computational Sciences, Massey University, Palmerston North 4410, New Zealand; 3School of Sport Exercise and Nutrition, College of Health, Massey University, Auckland 0632, New Zealand; p.r.vonhurst@massey.ac.nz

**Keywords:** vegan diets, amino acids, protein intake, protein digestibility, protein quality, meal pattern, data driven, hierarchical clustering

## Abstract

**Background/Objectives:** Inadequate intake of indispensable amino acids (IAAs) is a significant challenge in vegan diets. Since IAAs are not produced or stored over long durations in the human body, regular and balanced dietary protein consumption throughout the day is essential for metabolic function. The objective of this study is to investigate the variation in protein and IAA intake across 24 h among New Zealand vegans with time-series clustering, using Dynamic Time Warping (DTW). **Methods:** This data-driven approach objectively categorised vegan dietary data into distinct clusters for protein intake and protein quality analysis. **Results:** Total protein consumed per eating occasion (EO) was 11.1 g, with 93.5% of the cohort falling below the minimal threshold of 20 g of protein per EO. The mean protein intake for each EO in cluster 1 was 6.5 g, cluster 2 was 11.4 g and only cluster 3 was near the threshold at 19.0 g. IAA intake was highest in cluster 3, with lysine and leucine being 3× higher in cluster 3 than cluster 1. All EOs in cluster 1 were below the reference protein intake relative to body weight, closely followed by cluster 2 (91.5%), while cluster 3 comparatively had the lowest EOs under this reference (31.9%). **Conclusions:** DTW produced three distinct dietary patterns in the vegan cohort. Further exploration of plant protein combinations could inform recommendations to optimise protein quality in vegan diets.

## 1. Introduction

Dietary protein ingestion and its subsequent digestion within the gastrointestinal tract is crucial to the provision of indispensable amino acids (IAAs) which cannot be endogenously supplied by the human body [1,2,3]. IAAs are mostly absorbed through the ileal membrane into the bloodstream and then utilised for protein synthesis, the maintenance of net protein balance, and other metabolic functions [3,4]. One unique trigger for muscle protein synthesis (MPS) is the postprandial elevation of IAAs, particularly leucine, in the plasma and intracellular spaces, and this needs to be achieved through protein consumption [1,5,6]. Plasma amino acids are absorbed by body tissues such as the skeletal muscle until metabolic capacity (i.e., “muscle-full effect”) is achieved [6,7]. The synchronous supply of all IAAs within a single eating occasion (EO) is essential, as the effectiveness of protein synthesis may be constrained by the least available amino acid [8]. As the body is incapable of storing excess amino acids for later metabolic function, the simultaneous presence of all IAAs within a limited timeframe underscores the importance of carefully balancing dietary protein consumption within each EO [9,10].

Comparatively, the number of individuals who adopt a vegan diet (including only foods of plant origin) is relatively lower than the omnivorous diet and other PB diets such as the vegetarian diet. This is observed among European countries, as well as in New Zealand (NZ) [11,12]. Recently, the NZ Health Survey reported that only 0.74% of 23,292 surveyed respondents were vegans [11]. However, globally, the interest in vegan diets is increasing especially among younger adults [12,13]. This marks a pivotal point to critically examine the adequacy of such restrictive dietary patterns, given that significant nutrient deficiencies can occur in poorly planned vegan diets [14]. Among these, protein quality is a commonly overlooked aspect in these diets. Foods vary in AA composition, and this is especially true in plant-based (PB) foods, with many containing low concentrations of certain IAAs and thus considered to have low protein quality [9]. Given the crucial metabolic roles of each IAA in the human body, analysing the protein and IAA intake provides key information on overall adequacy across the day.

Foods in the human diet are rarely consumed in isolation [15], so combining an animal-sourced food (ASF) of higher protein quality is an efficient method to raise the overall IAA composition and digestibility of a mixed meal. This is, however, not an option in a vegan diet, which excludes all ASFs. Combining complementary PB foods with different limiting IAAs within each EO is therefore an efficient method for the simultaneous delivery of all IAAs to achieve threshold levels necessary for protein utilisation in a vegan diet. Understanding the current temporal distributions of dietary protein and IAAs, and how food combinations vary within such contexts are the first steps in identifying practical meal guidelines to improve overall protein intake and protein quality in vegan diets.

The protein adequacy of diets is currently considered by comparing total protein and IAA intake with their respective daily requirement values [16]. However, when protein adequacy is examined on a per-day instead of per-EO perspective, misleading conclusions about meeting protein requirements may occur [17]. The distribution and protein quality across meals of the day are masked with such an analysis, and thus, cannot account for how the evenness of protein distribution over time can impact physiological functions [18]. A high daily protein intake is the most significant factor in impacting lean body mass, but the combination of total protein quantity, protein quality, and meal (and snack) distribution may become pertinent in restrictive diets or more vulnerable populations [1].

Research on meal-pattern analysis for protein distribution is often hampered by the lack of well-defined per-meal requirements of total protein and IAA relative to body requirements. The current recommendation by a few experts for absolute protein intake per meal in a three-meal-per-day scenario is between 20 and 30 g to achieve metabolic benefits [19,20,21], but the target population in these studies varies, so it is unclear if this recommendation applies to the general adult population.

Past studies assessing the nutritional quality of meals have used time-of-day approaches, with meals generally designated as breakfast, lunch, and dinner [18,22,23]. Such methods are problematic, as pre-defined time windows for meals are inconsistent across studies and subject to researcher and classification bias—often associated with personal and social constructs—as compared to more novel data-driven methods [24,25,26]. Conventional techniques are also unlikely to capture meals outside of these time windows (such as for shift workers) and are open to debate on whether an EO should be classified as a meal or snack, depending on energy, nutritional content, or the situational context [15,27]. A more objective methodology is necessary to examine high-dimensional dietary and nutrition data [15,28]. Data-driven techniques allow for the detection of patterns in the diet independent of time of day [29] that may otherwise be missed with traditional approaches. It is then possible to assess whether groups of individuals with similar temporal dietary patterns have the same nutritional and health outcomes.

The aim of this study was to evaluate protein and IAA intake patterns among a sample of New Zealand (NZ) vegans and identify the temporal variation in total protein and IAA distribution across 24 h. We utilised Dynamic Time Warping (DTW) and hierarchical clustering (HC), two unsupervised methods in data analytics, to assess and classify distinct protein and IAA consumption patterns. A secondary aim was to investigate PB food combinations that emerged from the categorised dietary patterns. This study thus offers quantitative insights into how varied dietary patterns can affect protein metabolism and consequently inform future interventions to improve vegan meals of poorer protein intake and quality.

## 2. Materials and Methods

### 2.1. Study Design and Data Collection

Dietary data from 193 participants used in this study was obtained from the Vegan Health Research Programme, a cross-sectional study that examined the associations between vegan dietary patterns and nutrition and health outcomes among NZ vegans. Convenience sampling was used and healthy adults of at least 18 years who live in NZ were recruited through university advertisements, local vegan organisations, and word of mouth from June 2022 to August 2023. Our cohort comprised 73.1% females with an average age of 39 for females and 41 for males. The complete demographic profile of our population can be found in our previous study [30]. Considering that the vegan population in NZ constituted less than 1% of the surveyed population with the majority being female [11], we considered our sample size to be adequately representative of the demography of vegans in NZ.

Ethical approval was granted by the Health and Disability Ethics Committee (HDEC 2022 EXP 12312). All participants provided informed and written consent prior to data collection. Participants visited the Human Nutrition Research Unit at Massey University, Auckland, NZ on one occasion. As part of the inclusion criterion, only healthy adults 18 years and above who had followed a vegan diet for at least two years were recruited. Pregnant and lactating individuals were excluded. Participants were instructed to provide a detailed four-day food diary of all food, beverage, and supplement consumption, with a time and date stamp for four consecutive days.

### 2.2. Data Processing

Dietary data were processed using the FoodWorks Professional nutrient analysis software (Xyris, Brisbane, Australia, 2022) that is based on FOODfiles, the reference food composition table that provides nutrient content for foods in NZ [31]. The protein quality of a food refers to its capacity to supply utilisable amino acids to meet the body requirements of a specific population [9,32]. Hence, protein quality measurements must account firstly for overall AA composition and secondly for the digestibility of each AA in the food, the latter being an impactful factor that determines the utilisable proportion of each AA [33]. As AA composition data are not currently available from the NZ food composition tables, protein and IAA composition was matched to the most similar food item found in the food composition data from the United States Department of Agriculture (USDA) [34]. This was then normalised to the protein content of reported foods from NZ to approximate the IAA quantity in NZ foods. Total protein and each IAA were adjusted for digestibility using True Ileal Digestibility (TID) values from the available literature [35,36,37]. These values are currently available for tryptophan, threonine, leucine, lysine, methionine, cystine, and histidine. A full description of the data-processing method can be found in our previous study [30].

Date and time stamps from the processed dietary data were examined for accuracy for all EOs of each participant. When possible, missing date or time records were imputed by cross-checking with actual food records provided by participants. When manual imputation was not possible or accurate, the food records for that entire day were excluded. Six individuals with unclear or missing records were excluded. Time of food intake was absent in one individual for the first day. For the remaining five individuals, records for the 4th day were absent. Total protein and IAA quantities, adjusted for digestibility, were summed at each EO for each day and each participant. EO is defined as the time point where all foods were consumed together. Food items with less than 0.5 g of total protein per EO were excluded to reduce the number of food items for analysis. The remaining data still captured more than 98% of the total cohort’s protein intake and at least 92% of protein intake for every participant.

### 2.3. Data Analysis

DTW and subsequent clustering of time-series data were used to uncover patterns in the data and group them based on similarity [28]. The temporal distribution of intake patterns within the vegan cohort and how protein and IAA quantity vary as a function of time can thus be objectively categorised and subsequently examined [38]. All data-processing numerical analyses were conducted using R (version 4.3.1).

Unlike the well-established daily requirement patterns for protein and IAA [16], there are no well-defined reference or requirement values for protein and IAA intake per EO. Previous research provided values between 20 and 30 g of protein per meal or values of 0.24 to 0.4 g/kg of body weight (BW) [23,39,40], but varied depending on the protein source, population tested, and parameters measured (e.g., lean mass or muscle strength). However, the intake levels in this study cohort need to be compared to a suitable base measurement to evaluate the protein adequacy of a vegan diet throughout the day, and must especially be adjusted for body weight, which is the key factor for MPS [41]. The lowest possible threshold values obtained from the aforementioned studies were thus selected: 20 g per meal and 0.24 g/kg BW. Mean protein intake per EO for each individual was compared to this threshold, which was established as the minimal threshold to stimulate MPS [26,42,43]. This was justified by a significant increase in leucine oxidation above 20 g suggesting a “muscle-full effect” and no further protein anabolism above this threshold [42]. The percentage of individuals in the cohort with mean EOs below these threshold values was calculated as an indication of adequacy at the EO level.

Total protein intake at each EO across the day represented the data points within one time series for each individual. We denote this *individual.day* to represent the combinations of EOs per day for each participant. To group similar time series, we used a similarity measure calculated by DTW [38] with the dtw package in R [44]. Briefly, this approach obtains a warping path of minimum distance between two time series to align patterns of the same size [45,46,47]. To account for the varying lengths of time-series sequences, the normalised distance was used. Using total protein consumed per EO as the comparison parameter, a distance matrix was tabulated that captured the similarity between participants’ daily protein patterns (*individual.day*) across all four days. Lower DTW values indicated greater similarity between time series, showing more similar protein consumption patterns. Self-comparison was avoided by ensuring that DTW calculated distance between distinct *individual.days*. To validate the similarity measure calculated by DTW, several time series were selected from each cluster and the temporal protein distribution pattern was mapped and compared with the computed pairwise distance.

Figure 1 and Table 1 below provide a visual representation of a subset of participants and an example of how protein intake patterns were validated against the distance matrix computed by DTW. For example, 2.1 and 2.2 represent the same individual’s (individual 2) dietary pattern over two days (day 1 and day 2, respectively). These samples were more similar to each other than to 4.2 (individual 4 on day 2); their pairwise distance was smaller (2.34), compared to that between 2.1 and 4.2 (25.0). This was similarly observed when comparing individual 2 to individual 3. Sample 5.3 (individual 5 on day 3) was more similar to both 2.1 and 2.2 than it was to 4.2. By warping their time axes, the DTW method computed a minimal distance between two time series. This computed distance did not consider the duration between time points where the peaks occurred (for, e.g., differences in time of first meal), or the duration of each EO (how long foods were consumed), but aligned the overall “shape” of dietary intake between each *individual.day*.

Each *individual.day* was placed as a data object in a dendrogram with the agglomerative method of HC [48], which grouped these objects based on their similarity as calculated using DTW. Ward’s minimum variance method was applied to merge clusters that have the smallest increase in the value of the sum-of-squares variance [47]. The output of the HC is visualised in the dendrogram in Figure 2, which was enhanced by colouring branches to distinguish clusters, with the number of clusters set to *k* = 3. The appropriate number of clusters (*k*) was indicated by domain knowledge and guided by an average silhouette width (Figure A1). This is the optimal ratio that maximises inter-cluster distance and minimises intra-cluster distance, thus providing an evaluation of clustering validity [49].

To further visualise the clusters, the DTW distance matrix was projected into a two-dimensional space using multidimensional scaling (MDS) [50]. This step transforms the DTW distance matrix into coordinates in a 2D plane to interpret cluster separation and represent the relative distances between participants based on patterns of EOs.

### 2.4. Cluster Analysis

Each cluster represented collections of *individual.days* with similar patterns for protein intake. To investigate how the DTW algorithm identified similarities between time series, clusters derived from the computed distance matrix were analysed for variations in protein quantity, IAA composition, frequency of EOs and food-group contribution.

#### 2.4.1. Variation Across Clusters

To visualise trends in protein intake and frequency of EO across the two MDS dimensions (dim 1 and dim 2), mean protein (g), individual IAA intake (g) per EO and number of EOs per *individual.day* were calculated. The percentage of EOs below the stipulated protein reference values was calculated for each cluster. The temporal distribution of protein and individual IAA intake was evaluated across time segments of the day.

The contribution of individual food groups to protein and IAA intake was evaluated for each cluster. This step allows for the determination of how the composition and proportion of PB protein sources differ between clusters and their influence on protein and IAA intake. Each food item from the four-day food diary was matched to its respective food group, as defined in the 2008/2009 NZ Adult Nutrition Survey [51]. For instance, wheat and cereal-based foods were categorised as “grains and pasta” while beans, peas, and soy products were categorised as “legumes and pulses”. Food-group contribution for novel PB alternatives were also analysed to determine how clusters differed in terms of the consumption patterns for these foods. The principal protein-contributing ingredient in novel PB meat alternatives (PBMAs) and PB beverages (PBBs) was identified and matched to its respective food group.

#### 2.4.2. Statistical Analysis

To assess differences in protein and individual IAA intake, as well as body composition across clusters, and to identify which food groups exhibited significant differences in nutrient contributions among clusters, the Kruskal–Wallis (KW) test was used. A significant difference was identified when the *p*-value was <0.05. Subsequent pair-wise comparisons were conducted with the post hoc Dunn’s test with Bonferroni adjustments to determine the specific groups contributing to observed significant differences.

#### 2.4.3. Temporal Variation in Each Cluster

To capture the fluctuation of dietary behaviour over time and across different clusters, the time data were transformed into 24 hourly intervals, from 00:00 to 23:59. Total protein intake per EO was computed for each time segment and dietary cluster. To establish how protein quality differs across clusters, heat maps were created to visualise the distribution of mean IAA intake over hourly time segments across the different clusters.

## 3. Results

### 3.1. Total Protein Distribution of the Whole Cohort over 24 h

A total of 193 individuals who have completed dietary intake for at least three days were included in the temporal analysis of protein intake across the day. More than 80% of the population were New Zealanders of European descent and more than 90% of the cohort had completed tertiary education. Distinct protein intake patterns were observed across the cohort and the distribution of protein was also uneven throughout the day.

Figure 3A presents the overall distribution pattern for protein intake (y-axis) for all EOs (whose time is depicted on the x-axis) for all 193 participants across the four days of dietary recall. A high density of EOs contain protein content lower than 10 g. Fewer EOs contained 10 to 20 g and very few had >30 g of protein. These findings were corroborated by the histogram illustrated in Figure 3B, which showed a gradual decrease in density as protein intake increased. The highest density was observed in the lowest protein intake bin of <5 g per EO. Less than 2.5% of EOs contained protein above 40 g and were omitted in Figure 1, but can be observed from Figure A2. As a cohort, a mean intake of 11.1 g (standard deviation, sd = 5.3 g) of protein was consumed per EO and 93.5% of EOs contained <20 g of protein. Figure 3A and the histogram illustrated in Figure 3C showed three main peaks for total protein intake, which occurred at approximate time intervals of 07:00 to 08:00, 12:00 to 14:00, and 18:00 to 20:00 h. Total protein intake per EO was slightly higher in the later hours of the day as compared to the morning.

### 3.2. Cluster Analysis: Variation in Intake Patterns

A total of 766 time series (193 individuals × 4 days, with 6 *individual.days* excluded due to incomplete time records) were tabulated. Each observation point (or time series) was placed in its respective cluster based on the calculated DTW distance matrix (Figure 2). The agglomerative clustering stopped at a height of 60, when all observation clusters were merged. The results from the silhouette analysis (Figure A1) found the optimal number of clusters to be *k* = 2, as determined by the largest average silhouette width. However, as we observed two large clusters sharing the same merging node in Figure 2, *k* = 3, three clusters were selected for greater differentiation of the dataset (Figure 2 and Figure A3).

The three clusters were visualised on an MDS plot (Figure 4). Each point represents an individual time series, which corresponds to a unique *individual.day*. Dim1 and Dim2 accounted for 59.4% and 11.5% of the variance in these time series, respectively. The emergence of clusters was based on normalised DTW distances between time-series data. Clusters were analysed to determine how mean protein intake, and number of EOs differed across both dim 1 and 2. *Individual.days* as described in Figure 1 and Table 1 are represented in the MDS plot (Figure 4A), and the distance exhibited between these points further validates the calculated distance metric from DTW.

Mean daily protein intake per cluster was tabulated to be above the Australia and NZ Estimated Average Requirement (EAR) for both males and females, taken to be 0.68 g/kg BW/d for males and 0.60 g/kg BW/d for females [52], but was lowest in cluster 1 (0.64 g/kg BW/d) compared to clusters 2 (0.88 g/kg BW/d) and 3 (1.3 g/kg BW/d). The mean absolute protein intake per EO for cluster 1 was also the lowest (6.5 g), followed by cluster 2 (11.4 g), and highest in cluster 3 (19.0 g).

When adjusted to individual BW, the lowest protein intake (g/kg BW) per EO was in cluster 1, at 0.10 g/kg BW, followed by cluster 2, at 0.17 g/kg BW. The mean protein intake per EO was highest in cluster 3, at 0.27 g/kg BW, and above the selected reference value of 0.24 g/kg BW. Consequently, no EO in cluster 1 was meeting the target of 0.24 g/kg BW while clusters 2 and 3 had 91.5% and 31.9% of EOs below the target, respectively. In tabulating the mean total of all IAA per EO, cluster 1 also exhibited the lowest quantity of total IAA, with a mean of 1.3 g (0.4) per EO, followed by cluster 2, at 2.2 g (0.7) and cluster 3, at 3.8 g (1.4).

The number of EOs per *individual.day* was similar across clusters but highest in cluster 1 and lowest in cluster 3 (cluster 1 mean = 6.5 EOs/day; cluster 2 = 5.2 EOs/day; cluster 3 = 5.0 EOs/day). A larger number of EOs were congregated towards the top of the clusters though the distribution down dim 2 was inconsistent (Figure 4C), which could again be a consequence of the smaller variance exhibited by dim 2. Despite the high EOs per day in cluster 1, the mean protein per EO remained low, whereas in cluster 3, there were fewer EOs across the day but with higher protein intake per EO (Figure 4C).

Differences in body compositions were explored across the clusters. The mean BMI of individuals across clusters was similar. Cluster 1 was 23.8 (3.3), cluster 2 was 24.0 (3.1), and cluster 3 was 24.2 (2.7) and were not significantly different (*p* = 0.19). Mean percentage body fat was however highest in cluster 1, at 32.0% (6.5), followed by cluster 2 at 30.0% (7.6) and cluster 3 at 25.4% (6.8) with a significant difference across all clusters, as demonstrated by the KW and pair-wise comparisons with a post hoc Dunn’s test (*p* < 0.05).

### 3.3. Cluster Analysis: Contribution of Food Groups to Total Protein and IAA Intake

Figure 5 shows the percentage contribution of each food group to the total intake of protein and each IAA across the entire dataset. “Legumes and pulses” provided the highest contribution to total protein and most IAAs, particularly for lysine, across all clusters but was higher in clusters 2 and 3 than cluster 1. “Grains and pasta” was the next largest contributing food group to total protein and IAA (Appendix B Table A1) and, in particular, contributed most to the sulphur-containing amino acids (cystine and methionine) in cluster 1 as compared to clusters 2 and 3.

The KW test showed that for “legumes and pulses”, and “grains and pasta”, significant differences in absolute intake were observed across most clusters for protein and all IAAs, but were not significantly different for most of the other food groups. Table A2 presents the *p*-values from the post hoc Dunn’s test for pairwise comparisons between clusters for all nutrients and food groups. Clusters 1 and 3 were significantly different in many nutrients across most food groups as compared to 1–2 and 2–3 (Table A2). Protein and IAA contributions from novel PB alternatives such as PBMA and PB beverages (PBB) were explored for each cluster (Figure A4). As compared to clusters 1 and 2, protein and IAA contributions for novel PB foods in cluster 3 were provided almost entirely by “legumes and pulses”.

### 3.4. Cluster Analysis: Temporal Variation in TID-Adjusted Total Protein and IAAs

Based on spline curves in Figure 6, cluster 3 showed more variation and as shown by the colour gradient of individual plot points, total protein intake per EO was slightly higher in clusters 2 and 3. A slightly higher intake of protein in the later hours of the day than the morning were observed, more so for clusters 2 and 3. Overall, cluster 1 portrays EOs with the lowest total protein, which were more evenly distributed throughout the day as compared to clusters 2 and 3.

Figure 7 shows that the lowest mean intake of all IAAs across the day was observed in cluster 1, followed by clusters 2 and 3. This was particularly true for threonine, lysine, and leucine. Mean IAA intake was slightly higher in the later part of the day as indicated by the colour scale. This is most apparent for clusters 2 and 3 where the highest intake for these IAAs was between 18:00 and 21:00 h.

## 4. Discussion

This study evaluated the temporal distribution of protein and IAA intake among a sample population of NZ vegans. Our findings showed that total protein intake is unevenly distributed across the day, with higher intake at later hours. In examining absolute protein intake as a cohort (Figure 3), most vegans consumed less than 20 g of total protein per EO. Using DTW and HC, we objectively categorised participants into three distinct clusters which varied in total protein intake, IAA intake, and EO frequency throughout the day. This application of DTW aligns with prior research that also used this method to identify unique temporal patterns in energy intake across clusters [24].

### 4.1. Cluster Analysis

Clusters 1 and 2 contained 38.4% and 46% of total *individual.days* respectively, while cluster 3 contained 15.7% *individual.days*. Only cluster 3 was close to the stipulated reference intake of 20 g and 0.24 g/kg of BW of mean protein intake per EO and had the highest intake of most IAAs. Leucine and lysine intakes were at least 3x higher in cluster 3 than in cluster 1 (Figure 7). When using *k* = 2 for HC, as indicated by the silhouette analysis, clusters 2 and 3 were combined (Figure A3), while differences between clusters 2 and 3 were captured when k = 3 (Figure 2). We observe overlaps between clusters 1 and 2 and clusters 2 and 3 (Figure 4), indicating some similarity in dietary patterns, but it is apparent that the DTW algorithm has distinguished different protein intake and protein quality patterns, which were most different between clusters 1 and 3.

We have shown in our previous results that in this vegan cohort, increasing protein intake is associated with increasing IAA intake, although with varying degrees of association strength as protein intake increased above 50 g [30] Assessing total protein intake on a per-day basis tends to overestimate protein adequacy, as IAAs provided by dietary protein sources can only be utilised within a short time window following ingestion [17,53]. Unlike carbohydrates, fats, and some micronutrients, IAAs cannot be stored in the body apart from in skeletal muscle, which serves as a short-term reservoir [1,53]. Our previous study found that the mean daily protein consumption in this vegan cohort was above the Estimated Average Requirement [30]. Despite the achievement of mean daily protein intake per kg of BW for each cluster, this vegan cohort may face a challenge in achieving the minimal threshold of 20 g or 0.24 g/kg BW of protein per EO, especially for cluster 1 individuals. Our results show the importance of further evaluating protein adequacy at the EO level, as a per-day analysis tends to overestimate protein adequacy. Our findings aligned with that of Cardon-Thomas et al. (2017), who found a sub-optimal supply of protein in each meal despite the total protein intake per day being above Recommended Dietary Intake (RDI) for older adults [18].

### 4.2. Temporal Distribution of Dietary Protein

In this study, even though protein intake in cluster 1 appeared to be more evenly distributed as compared to the other clusters (Figure 6), none were above the minimal threshold of 0.24 g/kg BW. This may resemble a “grazing” or “snacking” pattern, as similarly observed among Australian females [29]. Cluster 1 is thus observed to have frequent but small EOs that contain low quantities of protein spread throughout the day, with few individuals achieving protein threshold levels. Cluster 3 on the other hand, had higher absolute protein intake and protein adjusted to body weight, at lower EO frequency. This raises the question of whether it is more effective to have protein intake throughout the day at a higher frequency in a grazing pattern but below the required threshold, or to have less frequent meals that meet the protein threshold.

An even distribution of protein intake throughout the day, as compared to a skewed pattern, was found to have potential benefits in physiological health, such as in MPS [18,20]. This, however, requires that each EO meets the minimal threshold for the target population. In opposition to this, a study by Arnal et al. (1999) found a positive impact on MPS in older adults using a pulse feeding strategy as compared to an even distribution of protein across the day [54]. Bollwein et al. (2013), however, stressed that each meal must still contain a minimal threshold of absolute protein content of at least 20 g to potentially achieve such a metabolic benefit [26]. A grazing pattern consisting of many EOs below the threshold (as seen in cluster 1) could more negatively impact MPS or other health outcomes than when more EOs are meeting this threshold (clusters 2 and 3). A well-controlled study is required to test this hypothesis and validate previously asserted meal protein thresholds.

An unbalanced intake pattern is metabolically inefficient, with only EOs at one point of the day meeting or exceeding the threshold. The presence of any excess IAAs is futile in the presence of at least one limiting IAA, as excess IAAs are lost to oxidation instead of maximally utilised within body tissues [41]. There was no clear difference in temporal variation across the clusters in our study, which could be an effect of small sample size in each cluster. However, the trends in Figure 1 show that protein intake was higher at the later parts of the day as compared to the start of the day and consistent with several studies examining protein distribution across the day in various populations [18,22,23,55].

In Johnson et al. (2022), among healthy adult women (18 to 79 years old) who consumed at least 0.24 g/kg/period or 25 g of protein at least once in three meals, a positive association was observed with lean mass, upper-body strength, and lower-body endurance [23]. Two other studies that examined dietary protein distribution to MPS, lean mass and muscle performance also identified that more frequent and even consumption of approximately 30 g of protein per meal had the greatest association with leg lean mass, strength [10], and MPS stimulation [20]. Other studies have suggested that the optimal protein intake per meal is 0.24 g/kg of body weight (BW) for younger adult men and 0.4 g/kg of body weight for older adult men, based on a breakpoint analysis of MPS data [40,41]. These conclusions were, however, based on the ingestion of high-quality protein and an absence of middle-aged individuals between 38 and 64 years [23].

Hence, sufficient protein intake is linked to the maintenance of lean mass, not just on a daily basis, but also at a per-meal level [10]. The BMI between cluster 1 and the other clusters was not significantly different but body fat percentage was significantly different across all clusters, with cluster 1 having the highest body fat percentage. Cluster 3, where individuals had higher mean protein intake (g/kg BW) and the lowest percentage of EO below the threshold, was observed to have the lowest mean body fat percentage. Without considering other confounders such as physical activity levels and energy expenditure, it is challenging to identify an association between protein intake and temporal distribution on body composition. This finding was, however, similar to results from a cross-sectional study which found higher fat-free mass in adults whose protein intake was at least 0.24 g/kg BW in each of three meals across the day, although these meals were unevenly distributed [39]. While it was not the intention of this vegan cohort study to assess the relationship between protein intake and body composition, this may be an interesting avenue for future work.

Stable isotope techniques have concluded that IAA availability is the most important factor for MPS stimulation with a dose-dependent mechanism [56,57]. Individuals who consistently fall below the EO threshold for dietary protein may receive insufficient concentrations of IAAs to meet requirements, and face risks in depleting intramuscular stores of IAAs. Taken together, it would be ideal for all EOs to meet the protein threshold. If this objective is not achieved, then at least meeting the quantity for one EO per day is important to achieve MPS stimulation. Based on results from the ingestion of whey protein (a rapidly digested protein), the deprivation of IAA intake induces the catabolic process of skeletal muscle protein breakdown (MPB) around three hours postprandially, and MPB is likely more significant between an overnight fast to the first meal of the day when the body is in a catabolic state [1]. Hence, it is perhaps more important for the protein threshold to be met at the first meal.

However, slower-digesting proteins may induce the prolonged elevation of IAA concentration in plasma [42] and negate the risk of MPB during post-absorptive periods. Studies that found elevated leucine levels at four to five hours postprandially were based on isolated leucine supplementation [7,58]. Hence, the type of protein source may matter in its delivery and maintenance of IAAs in the body. The plasma response of leucine and other IAAs may differ in mixed diets of varying protein quality, as compared to isolated protein sources which may induce greater aminoacidemia [10], but these differences are not sufficiently examined in the literature.

### 4.3. Food Group Contribution to Dietary Protein

To distinguish how different food groups contribute towards protein and IAA intake across the clusters, the percentage of total intake to protein and each IAA was tabulated for each food group. Figure 6 and Appendix B Table A1 demonstrated that across all clusters, “legumes and pulses” was the major group contributing to total protein and most of the IAAs, such as lysine, leucine, threonine, and histidine. As supported by both Figure 6 and Appendix B Table A2, the contribution of “legumes and pulses” was significantly higher in clusters 2 and 3 than cluster 1. Other evaluations highlighted the importance of leguminous foods to total protein and lysine intake [34,59,60,61]. “Grains and pasta” was the next largest contributing food group, most highly to methionine and cystine in cluster 1, and significantly different between cluster 1 and 3. These results suggest the potential that individuals in cluster 1 may consume a larger proportion of “grains and pasta” than “legumes and pulses”, while the opposite is true for clusters 2 and 3.

In analysing the contribution of the main protein-containing ingredient from novel PB foods, cluster 3 showed “legumes and pulses” as the dominant food group, while a smaller proportion of “grains and pasta” was observed in cluster 1. Most of the PBMA consumed by the NZ vegan cohort were made from isolated soy and pea-protein concentrates and isolates. Furthermore, these isolated proteins were processed with techniques that could increase the availability of amino acids, as shown by the digestibility coefficients utilised in this study, as well as other protein quality measures [36,62]. Soy and pea-based PBBs have higher IAA quantities than those made from oat and rice, particularly for lysine, and are closer to meeting the Food and Agriculture Organization (FAO) IAA requirements [63,64]. Hence, it could be valuable to prioritise “legumes and pulses” as the primary protein-contributing ingredient if novel alternatives are included within a vegan diet, especially if lysine and leucine quantities must be increased. This must, however, account for overall consumer acceptability, the balance of other micronutrients, and non-nutritional factors such as cost.

### 4.4. Strengths and Limitations

This study has several limitations, the first of which has relevant impact on measuring protein adequacy at the meal level. The minimal threshold of protein per EO was based on literature values, but these values varied between 15 and 45 g of protein per EO across numerous studies and was dependent on the type of protein tested and the study population. The minimal threshold of 20 g of protein per EO taken in our study was based on findings from only six young men after resistance exercise, so the minimal threshold may be elevated in other populations, especially in sedentary older adults, who face the highest risks of protein deficiency [16]. Furthermore, this minimum quantity may be based on high-quality protein containing 10 g of all IAAs [65,66]. Higher levels may be warranted in diets providing lower concentrations of utilisable IAAs, such as in a vegan diet. Setting the minimum threshold at a conservative 20 g provided a benchmark to assess protein adequacy and avoid overstating inadequacy that could occur with a higher threshold. Well-defined threshold levels for protein and IAA intake would be valuable, especially for populations on restrictive diets or at more advanced ages.

This cross-sectional study was dependent on a four-day food diary, which offered a brief view of dietary behaviour that may not represent true temporal variations in nutrient intake for individuals [27]. Weekday and weekend differences in meal patterns were not considered but could be important given that various social and lifestyle behaviours may affect the type of food consumed at each EO and the number of EOs, and thus impact the results. However, the tabulation of the mean intake of nutrients per EO over four days was conducted to correct intra-individual variations during the food-recording period.

One strength of this study was the use of novel quantitative techniques to identify meal patterns objectively, rather than pre-define time points for meals and snacks. This is especially useful because definitions of meals and snacks are not clear in the literature and segregating consumption points based on conventional “breakfast, lunch and dinner” time patterns is subjected to bias and inter-individual variation. While some domain expertise was needed to decide the optimal number of clusters (*k*), the selection of three clusters was guided by silhouette analysis and validated by comparing to results from *k* = 2.

The mechanism behind DTW is the computation of mean distance based on alignment of time series of different lengths [67]. However, DTW does not distinguish time series based on daily consumption duration or the specific timing of meal initiation in the day. Consequently, the analysis may not capture differences in daily meal duration or starting points across individuals.

Another strength in this study was the adjustments for TID for both protein and individual IAAs. While other factors apart from digestibility can impact protein and IAA metabolism in the human body, digestibility adjustments taken at the terminal ileum serve as a valid proxy for the quantity that can be utilised by the body [33]. This is an important step in the evaluation of how adequately a vegan diet can provide for high-quality protein since plant foods are known to have lower protein digestibility due to anti-nutritional compounds and inherent structural configurations within their protein structures. While normalising IAA quantities from matched foods in the USDA database to those consumed in this cohort provides a representation of actual intake, the degree of accuracy in this normalisation remains uncertain.

### 4.5. Implications for Future Research

The emergence of dietary patterns that varied in protein intake and protein quality in this study tells us that certain consumption patterns of a vegan diet may be more effective in increasing the daily protein intake and protein quality to requirement levels. This is observed especially in cluster 3 where individuals had higher total protein and IAA intake in EOs across the day while maintaining lower frequency of meal intakes. Balanced intake of diverse high-quality plant protein sources in meals or snacks within narrow time windows could be the key to achieving protein adequacy in vegan diets while keeping to energy requirements. Meals and snacks found in the dietary data of cluster 3 could provide information on healthy vegan diets that meet protein requirements and be adapted to dietary patterns in clusters 1 and 2. This is worth exploring and is the focus in our future research.

## 5. Conclusions

This study highlights the impact of dietary patterns on protein and IAA adequacy within a NZ vegan population through novel quantitative techniques to objectively categorise distinct variations. DTW distinguished three dietary patterns in this vegan cohort, with cluster 3 being most different to cluster 1. Cluster 1 displayed markedly lower total protein and IAA intake and compared poorly to the stipulated absolute and relative protein reference values. Individuals who follow dietary patterns in cluster 1 may be most at risk for protein deficiency, especially if these daily diets are followed over longer periods. A larger proportion of “legumes and pulses” at the EO level for cluster 1 may improve total protein and IAA intake. Our study also corroborates earlier research that protein consumption tends to peak in the later hours of the day regardless of cluster assignment, raising concerns about protein distribution and metabolic inefficiencies over the day with potential negative impacts on lean muscle mass and MPS stimulation.

Plant-sourced foods often have lower concentrations of digestible IAAs that can match up to individual requirements as compared to animal-sourced foods. Identifying appropriate PB protein sources in the right proportions is a practical way to achieve combinations of foods that can complement the limiting IAAs and improve the protein adequacy of a vegan diet. This complementary effect only happens in the same EO and must occur within a specified time window. Every EO is thus an opportunity for optimal nutrient intake. Our results showed that improvements to the diet in cluster 1 may be possible by utilising food combinations from cluster 3, but this approach must be balanced for overall energy, fibre, and other nutrients, an important step towards optimising protein adequacy for the vegan population.

## Figures and Tables

**Figure 1 nutrients-17-01806-f001:**
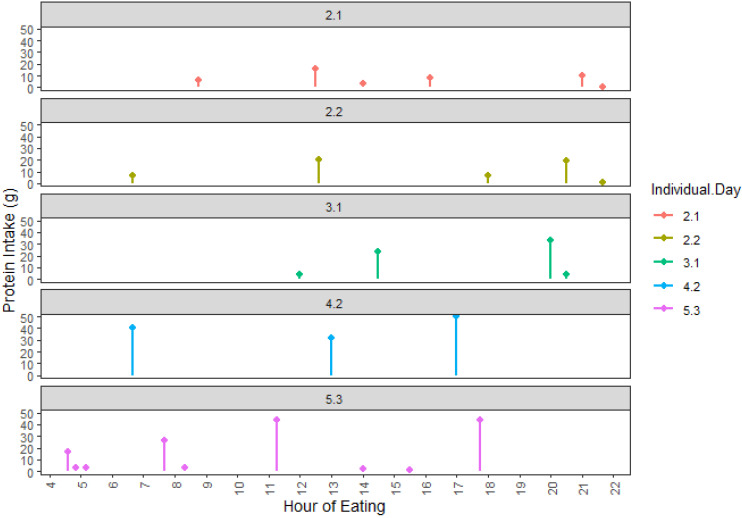
Protein intake as a function of time of the day for several sample *individual.day*s. 2.1 represents individual 2′s protein intake pattern for day 1, and 2.2 represents the same individual on day 2. 3.1 represents individual 3’s protein intake pattern for day 3. 4.2 represents individual 4’s protein intake pattern on day 2 and 5.3 represents individual 5’s protein intake pattern for day 3.

**Figure 2 nutrients-17-01806-f002:**
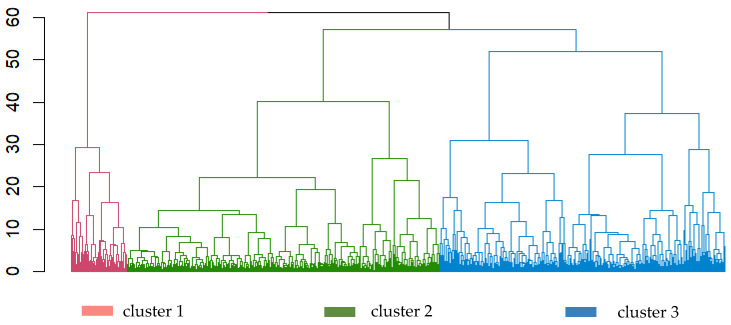
Dendrogram clusters for each *individual.day* (total of 766 time series/observations) using hierarchical clustering with Ward.d2 method [48]. Using the agglomerative method, each data point is merged to form a cluster, and merging continues until a single data point is formed, at height 60. Vertical lines represent the branches, and horizontal lines represent fusions of clusters or data points. The longer the vertical line, the larger the height, and the more dissimilar the data points or clusters.

**Figure 3 nutrients-17-01806-f003:**
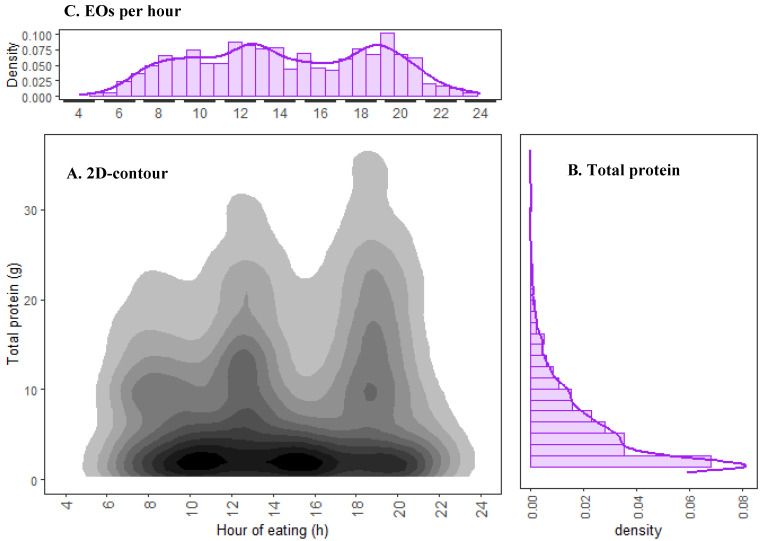
Two-dimensional contour estimate (**A**) showing the temporal distribution of total protein (g) per EO for all 193 participants over four days. The varying density of total protein intake is represented by a grey colour ramp; darker areas have more EOs. Histograms (purple) represent density of total protein consumed (g) (**B**) and density of EOs per hour of eating (**C**). The temporal distribution of protein intake per EO showed that the vast majority of EOs occurred between 04:00 and 24:00. In total, <10 EOs were observed outside of these hours (presented in Figure A2).

**Figure 4 nutrients-17-01806-f004:**
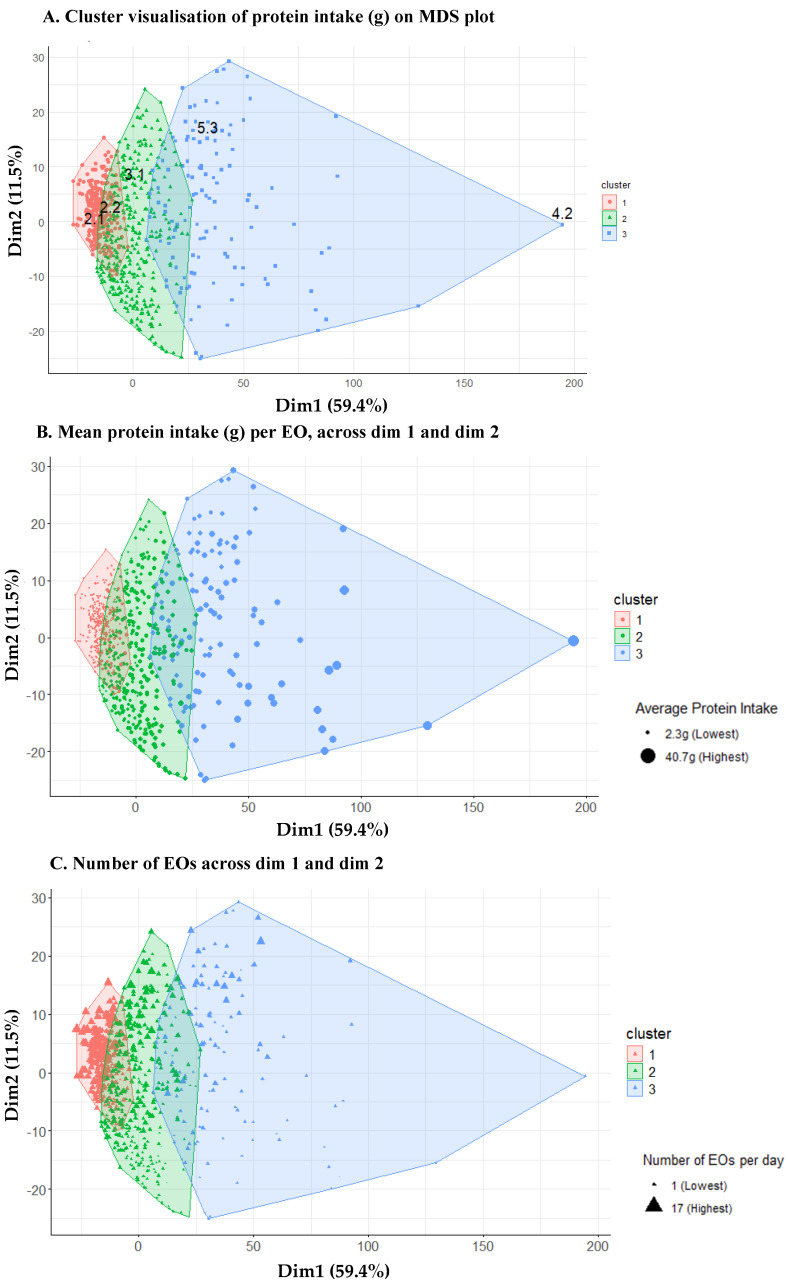
Cluster visualisation of the mean protein intake pattern per EO for each *individual.day* on the MDS plot. Convex envelopes as shown by coloured lines around each cluster represent the boundaries of each cluster. Examples of *individual.day* described in Figure 1 and Table 1 are depicted (**A**). A cluster visualisation of how mean protein intake changes across clusters (**B**). The larger the point (shown as a dot), the higher the quantity of protein. The size of dots is the largest in cluster 3, and smallest in cluster 1. Mean protein intake increases towards the right of dim 1. Cluster visualisation in which each point on the MDS plot represents the number of EO for each *individual.day* (**C**). The larger the point (shown as a triangle), the higher the number of EOs. The size of points are smallest in cluster 3, and larger in cluster 1, as well as larger at the top of dim 2. This indicates that the frequency of EOs decreases across dim 1 and down dim 2.

**Figure 5 nutrients-17-01806-f005:**
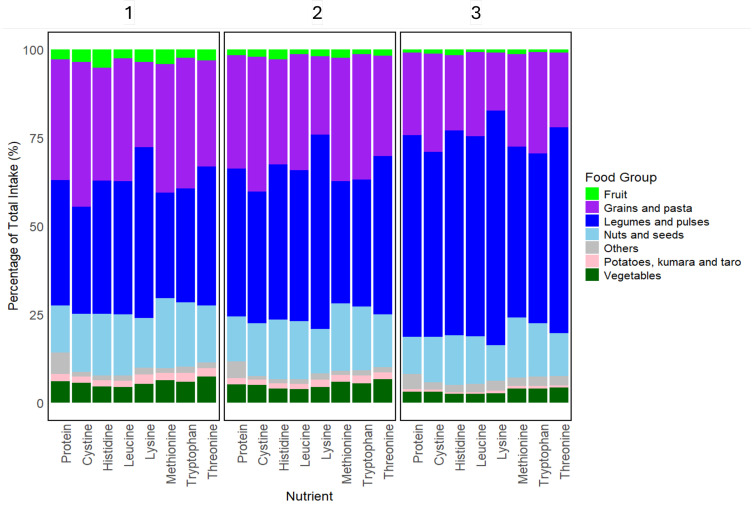
The contribution of food groups to TID-adjusted protein and IAA intake, per cluster (numbered 1 to 3), as a percentage of total intake. “Others” included “beverages”, “savoury sauces and condiments”, and “sugar and sweets” that each contributed <5% of total protein.

**Figure 6 nutrients-17-01806-f006:**
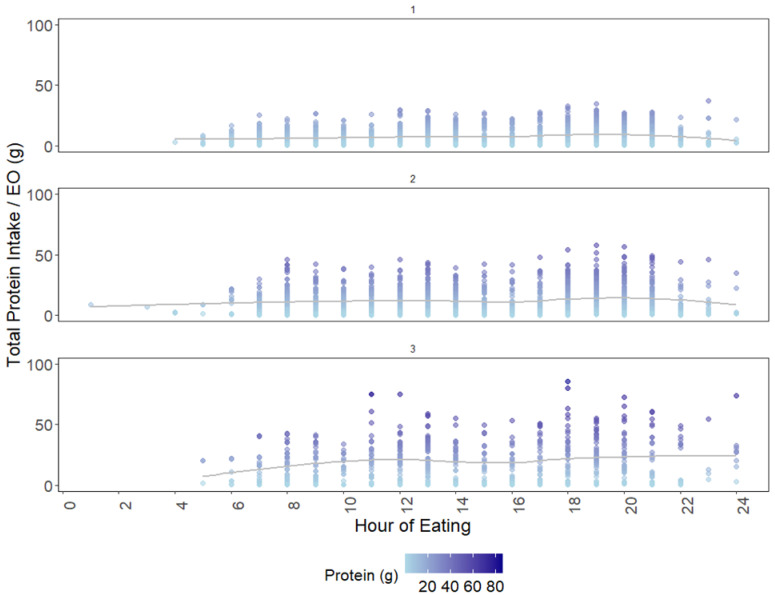
Temporal patterns showing the distribution of protein intake over 24 h of the day for all *individual.days* in each cluster (numbered 1 to 3). Points represent total protein intake (g) from each EO. Grey line indicates a spline curve to outline the relative trend over time.

**Figure 7 nutrients-17-01806-f007:**
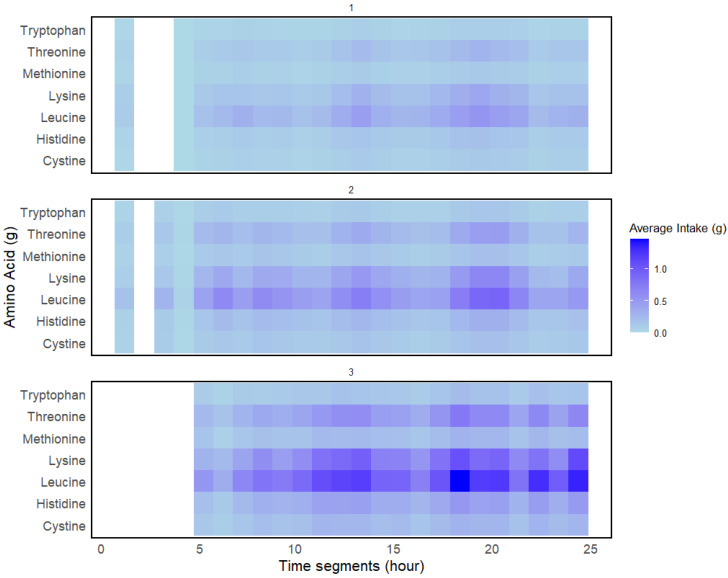
Distribution of mean IAA intake (g) per EO across 24 h time segments for clusters 1 to 3 (numbered). A gradient colour scale was used where darker shades indicate higher intake.

**Table 1 nutrients-17-01806-t001:** The sample representation of a two-dimensional distance matrix of pairwise normalised distance as computed using DTW. Each participant’s unique protein intake distribution across the day was depicted as one time series (*individual.day*). The application of DTW compared and matched the patterns of two time series. For example, 2.34 is the resulting distance between the *individual.days*, 2.1 and 2.2. This is smaller than the distance between *individual.days* 2.1 and 3.1, which is 5.45, indicating greater similarity between 2.1 and 2.2 than either is to 3.1. Based on the calculated distance, *individual.day* 4.2 differs most from the other time-series patterns.

	2.1	2.2	3.1	4.2
**2.2**	2.34			
**3.1**	5.45	5.70		
**4.2**	25.0	21.9	14.0	
**5.3**	8.07	7.99	8.44	19.6

## Data Availability

All supporting data and accompanying codes for the generation of reported results are available in the Appendix A.

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
