# Peer review of "Protein Intake and Protein Quality Patterns in New Zealand Vegan Diets: An Observational Analysis Using Dynamic Time Warping"

_nutrients, 2025, doi:10.3390/nu17111806_

Round 1
Reviewer 1 Report
Comments and Suggestions for Authors
Dear Authors,
Thank you for addressing this important topic with your submitted manuscript. I believe your study aligns well with the journal's objectives, and I have included my comments and suggestions below:
Title: Please include the study design in the title.
Introduction:
I find your introduction to be clear and well-written. However, I would ask you to expand it by adding a paragraph that provides a more international overview, which can then lead into the context of your own country. This could focus on the growing trend of dietary changes, such as the increasing choice to adopt vegan or plant-based diets, etc. For example, several recent studies in the European landscape have explored the modification of the Mediterranean diet with the adoption of plant-based diets, such as the "comparison between plant-based diets and the Mediterranean dietary pattern and their socio-demographic determinants in the Spanish population". Following this, you could better focus the paragraph on your specific context.
Methods:
This section is excellently written and supported by high-quality tables and figures. To further strengthen the methodology, which I already consider solid, I would suggest referring to the Equator Network Guidelines in the first section of the Methods.
Results and Discussion:
I would like to congratulate the authors for the quality of your manuscript; the text is clear and well-written.
I believe the graphical quality of the images is adequate.
To further enhance the quality of your manuscript, I suggest adding a final section before the conclusions that addresses the future research implications of your study.
References:
Please ensure the references follow the journal’s guidelines.
Author Response
Comment 1:
- Thank you for addressing this important topic with your submitted manuscript. I believe your study aligns well with the journal's objectives, and I have included my comments and suggestions below:
Title: Please include the study design in the title.
Response 1: Thank you for the feedback and suggestion to include the study design in the title for improved clarity. We have added the term “observational study” to the title on page 1, lines 1 to 4.
Comment 2: I find your introduction to be clear and well-written. However, I would ask you to expand it by adding a paragraph that provides a more international overview, which can then lead into the context of your own country. This could focus on the growing trend of dietary changes, such as the increasing choice to adopt vegan or plant-based diets, etc. For example, several recent studies in the European landscape have explored the modification of the Mediterranean diet with the adoption of plant-based diets, such as the "comparison between plant-based diets and the Mediterranean dietary pattern and their socio-demographic determinants in the Spanish population". Following this, you could better focus the paragraph on your specific context.
Response 2: Thank you for this suggestion which will help to set the appropriate context where protein quality is an important aspect to address in vegan diets. We also appreciate the provision of a relevant study that compared plant-based diet adoption to Mediterranean dietary patterns and have included this as part of the introduction. An additional paragraph providing context for our study has been added in page 2, lines 45 to 57.
Comment 3: This section is excellently written and supported by high-quality tables and figures. To further strengthen the methodology, which I already consider solid, I would suggest referring to the Equator Network Guidelines in the first section of the Methods.
Response 3: We have provided an additional paragraph in the first section of the methods to describe the population sampled, following guidelines from observational studies found at the Equator Network Guidelines. Thank you for the suggestion. Using the STROBE guidelines for methods, we have added information pertaining to the setting of the study on page 3, lines 106-123.
Comment 4: I would like to congratulate the authors for the quality of your manuscript; the text is clear and well-written.
I believe the graphical quality of the images is adequate.
To further enhance the quality of your manuscript, I suggest adding a final section before the conclusions that addresses the future research implications of your study.
Response 4: We appreciate the suggestion to relate our findings to future implications that we hope to address in the next phase of our research. We have added a section describing the important findings and their implications in our future research in page 17, lines 574 to 584.
Comment 5: Please ensure the references follow the journal’s guidelines.
Response 5: Guidelines have been checked and follow the journal guidelines
Reviewer 2 Report
Comments and Suggestions for Authors
- Line 29: The abbreviation “IAA” needs to be introduced in the main text.
- Methods: Please mention the total number of study participants already in the methods section. Please also elaborate in the recruitment procedure.
- Please also elaborate whether the sample is representative for vegans (in New Zealand).
- Hoy many missing values? And how did you deal with them?
- Results: Please add a sample description.
- Line 250: What is meant by “on the right”?
- Line 297: What does the value “48” relate to?
- Line 304: Please write “Cluster 1” instead of “Cluster one”.
- Line 346f.: There seems to be a formatting issue. Please check.
- Figure 7: Please explicitly mention that the values 1-3 refer to the respective Cluster.
Author Response
Comment 1 - Line 29: The abbreviation “IAA” needs to be introduced in the main text.
Response 1 - Thank you for pointing this out and the abbreviation has been included in page 1, line 29
Comment 2 - Methods: Please mention the total number of study participants already in the methods section. Please also elaborate in the recruitment procedure.
Response 2 - We have added more information about the recruitment procedures and number of study participants in the revised manuscript, in pages 2 to 3, lines 93 to 101.
Comment 3 - Please also elaborate whether the sample is representative for vegans (in New Zealand).
Response 3 - Thank you for this suggestion. We have added our input on the representation of our vegan cohort to the available data of NZ vegans, on page 3, lines 95 to 99.
Comment 4 - How many missing values? And how did you deal with them?
Response 4 - We have added this information to section 2.2 of Methods and the number of daily records removed from the analysis are explained on page 3, lines 131 to 133
Comment 5 - Results: Please add a sample description.
Response 5 - We have added a description about the demographic data of the sample at the start of the results section, and a brief statement on dietary protein patterns in the population. This can be found on page 7, lines 255 - 259
Comment 6 - Line 250: What is meant by “on the right”?
Response 6 - We have improved the descriptions of the figures. These can be found on page 7, line 264 and 269
Comment 7 - Line 297: What does the value “48” relate to?
Response 7 - This is a reference for the EAR values of protein for adult males and females, obtained from the FAO guidelines. We have changed the reference brackets for improved clarity
Comment 8 - Line 304: Please write “Cluster 1” instead of “Cluster one”.
Response 8 - We have addressed this on page 10, line 318
Comment 9 - Line 346f.: There seems to be a formatting issue. Please check.
Response 9 - Thank you for drawing our attention to this. We have reformatted figures 5 and 6
Comment 10 - Figure 7: Please explicitly mention that the values 1-3 refer to the respective Cluster.
Response 10 - We have corrected this in Figure 7, on page 13, line 374
Round 2
Reviewer 1 Report
Comments and Suggestions for Authors
The authors have provided appropriate modifications to the manuscript.
Reviewer 2 Report
Comments and Suggestions for Authors
All of my previous recommendations have adequately been addressed.